# Investigation of RNA Editing Sites within Bound Regions of RNA-Binding Proteins

**DOI:** 10.3390/ht8040019

**Published:** 2019-11-29

**Authors:** Tyler Weirick, Giuseppe Militello, Mohammed Rabiul Hosen, David John, Joseph B. Moore, Shizuka Uchida

**Affiliations:** 1Cardiovascular Innovation Institute, University of Louisville, Louisville, KY 40202, USA; 2RIKEN Center for Integrative Medical Sciences (IMS), 1-7-22 Suehiro-cho, Tsurumi-ku, Yokohama 230-0045, Japan; 3Department of Molecular Cellular and Developmental Biology, Yale University, Yale Science Building-260 Whitney Avenue, New Haven, CT 06511, USA; giuseppe.militello@yale.edu; 4Department of Internal Medicine-II, Molecular Cardiology, Biomedical Center (BMZ), University of Bonn, Sigmund-Freud-Str. 25, Bonn 53127, Germany; hosenmr@uni-bonn.de; 5Institute of Cardiovascular Regeneration, Centre for Molecular Medicine, Goethe University Frankfurt, Theodor-Stern-Kai 7, Frankfurt am Main 60590, Germany; John@med.uni-frankfurt.de; 6The Christina Lee Brown Envirome Institute, Department of Medicine, University of Louisville, Louisville, KY 40202, USA; joseph.moore@louisville.edu; 7Diabetes and Obesity Center, University of Louisville, Louisville, KY 40202, USA

**Keywords:** RNA, RNA editing, RNA-seq, RNA-binding proteins, transcriptome

## Abstract

Studies in epitranscriptomics indicate that RNA is modified by a variety of enzymes. Among these RNA modifications, adenosine to inosine (A-to-I) RNA editing occurs frequently in the mammalian transcriptome. These RNA editing sites can be detected directly from RNA sequencing (RNA-seq) data by examining nucleotide changes from adenosine (A) to guanine (G), which substitutes for inosine (I). However, a careful investigation of such nucleotide changes must be conducted to distinguish sequencing errors and genomic mutations from the genuine editing sites. Building upon our recent introduction of an easy-to-use bioinformatics tool, RNA Editor, to detect RNA editing events from RNA-seq data, we examined the extent by which RNA editing events affect the binding of RNA-binding proteins (RBP). Through employing bioinformatic techniques, we uncovered that RNA editing sites occur frequently in RBP-bound regions. Moreover, the presence of RNA editing sites are more frequent when RNA editing islands were examined, which are regions in which RNA editing sites are present in clusters. When the binding of one RBP, human antigen R [HuR; encoded by ELAV-like protein 1 (ELAV1)], was quantified experimentally, its binding was reduced upon silencing of the RNA editing enzyme adenosine deaminases acting on RNA (ADAR) compared to the control—suggesting that the presence of RNA editing islands influence HuR binding to its target regions. These data indicate RNA editing as an important mediator of RBP–RNA interactions—a mechanism which likely constitutes an additional mode of post-transcription gene regulation in biological systems.

## 1. Introduction

The life of RNA is more complex than previously thought. Upon transcription from DNA, RNA is modified by a variety of enzymes, resulting in over 140 types of RNA modifications [1]. These modifications are increasingly implicated in various pathophysiological conditions, findings that have emboldened the field of epitranscritomics [2,3,4] to rapidly expand and focus efforts on delineating the role of RNA modifications in disease etiology [5]. Among various RNA modifications, RNA editing is particularly interesting, as it can be detected directly from RNA sequencing (RNA-seq) data [6]. RNA editing is a post-transcriptional modification that alters the sequence of RNA molecules [7,8]. Differing from 5’-capping, 3’-polyadenylation, and splicing, RNA editing modifies RNA molecules by the insertion, deletion, or base substitution of nucleotides via RNA editing enzymes to diversify the resulting transcripts. When RNA editing takes place in exons of protein-coding genes, an amino acid sequence that is different from the original DNA sequence could arise, whereas editing of 3´-untranslated regions (UTRs) may affect binding of RNA binding proteins or microRNAs (miRNAs) to modulate RNA stability and/or translation [7]. 

There are two types of RNA editing: adenosine to inosine (A-to-I) and cytidine to uridine (C-to-U), where A-to-I is the most common form. A-to-I editing occurs through RNA editing enzymes, adenosine deaminases acting on RNA (ADARs), which convert adenosine (A) in double-stranded RNA into inosine (I). Although A-to-I editing occurs at the RNA level, when reverse transcribed to complementary DNA (cDNA), an inosine is converted to guanine (G). This A-to-G conversion, which is evidence for RNA editing sites, may be simply identified by comparing resultant cDNA sequences to the reference genome [9]. In mammals, there are three ADARs (*ADAR1*, *ADAR2*, and *ADAR3*). Although ADAR3 (also known as *Adarb2*—adenosine deaminase, RNA-specific, B2) is considered to be catalytically inactive and a less important player in RNA editing events [10], the knockout mice of *Adar1* and *Adar2* (also known as *Adarb1*—adenosine deaminase, RNA-specific, B1) are embryonically and postnatally lethal, respectively [11,12], which highlights the importance of RNA editing events in normal development. Given the importance of ADARs, a number of studies have conducted experiments to detect RNA editing events from RNA-seq data [13,14,15,16,17,18], including the one that we recently reported in endothelial cells [19]. The most challenging part of such identification is the separation of true RNA editing sites from genomic variations (e.g., single nucleotide polymorphisms (SNPs)), which are distinguished by a series of filters [20,21,22,23,24,25,26,27]. However, there is a lack of easy-to-use bioinformatics tools to analyze RNA editing events. To this end, we recently developed a new bioinformatics tool, RNAEditor [28], to assist the discovery of RNA editing events without extensive knowledge about programming.

Besides ADARs, there are many proteins that bind to RNAs. In terms of RNA binding, there is a protein domain called RNA recognition motif (RRM), which binds to single-stranded RNAs (https://www.ebi.ac.uk/interpro/entry/IPR000504). When a protein owns an RRM domain, it is generally categorized as an RNA-binding protein (RBP). Because many RBPs are pathophysiologically important [29,30,31,32,33,34,35], there is a growing interest in studying RBPs. These interests have stimulated the development of various experimental techniques, including several forms of RNA immunoprecipitation (RIP) followed by next generation sequencing (NGS) (called RIP-seq [36,37]). Such techniques include crosslinking-immunoprecipitation (CLIP) followed by NGS (CLIP-seq [38]; also known as high-throughput sequencing of RNA isolated by crosslinking immunoprecipitation (HITS-CLIP)), photoactivatable ribonucleoside-enhanced crosslinking and immunoprecipitation (PAR-CLIP) [39], and individual-nucleotide resolution UV cross-linking and immunoprecipitation (iCLIP) [40,41]. Essentially, all of these techniques are performed on a genome-wide scale to identify RNA target sequences that RBPs may bind to. To avoid confusion, in this study, all of these variations of RIP-seq are collectively called CLIP-seq. Furthermore, there are established databases to utilize the information obtained from individual laboratories to facilitate continued research efforts that seek to understand the consequence of binding of (or lack of) RBP. Such databases include CLIPdb [42], MitBase [43], POSTAR [44], and starBase [45,46]. These systematically collected and cataloged CLIP-seq data provide a stage for further bioinformatics analysis of RBPs and their interacting RNA consensus sequences.

Given that most RBPs recognize and bind in a sequence-specific manner (specific RNA-binding motifs), it is not unsurprising that mutations in such binding motifs (both in RBP and their bound RNA sequences) would alter RBP binding capacity. As A-to-I RNA editing occurs rather frequently in the transcriptome [28], it is not surprising that RNA editing alters the binding of RBP. These alterations include the changes in miRNA binding sites, which are loaded onto RNA-induced silencing complex (RISC) that include Argonaute (AGO) protein family as RBP [19,47,48,49,50]. Although bits and pieces of information are available, there is not a systematic investigation of RBP and their bound sequences that could be altered by RNA editing. Here, utilizing the available information and bioinformatics tools, we dissected how RNA editing sites alter the binding of RBP, which may have consequences on cellular physiology.

## 2. Materials and Methods

### 2.1. Data Sources

For CLIP-seq data, the analyzed data were downloaded from starBase [45,46]. For RNA editing sites, the analyzed data were downloaded from RADAR [27]. All of the above datasets were originally mapped to hg19. In order to avoid any biases resulting from further secondary data analysis, the data were downloaded from the above databases without any modifications. To identify the RNA editing sites within RBP-binding regions, the intersect command of BEDTools [51] with the “-wo” option was used.

To identify RNA editing sites and islands, the following RNA-seq data of HEK-293 cells were analyzed with RNAEditor [28]: SRR3994124, SRR3994125, SRR3994126, SRR629569, and SRR629570 (all are from the Sequence Read Archive (SRA)).

### 2.2. Culturing of Cells, qRT-PCR, and siRNAs

HEK-293 cells were cultured in growth medium consisting of Dulbecco’s Modified Eagle Medium (DMEM) with low glucose and pyruvate (Thermo Fischer Scientific, Darmstadt, Germany) supplemented with 10% fetal bovine serum (FBS) (Thermo Fischer Scientific, Darmstadt, Germany), and antibiotics (100 units of penicillin and 100 μg of streptomycin per milliliter, Sigma-Aldrich, Hamburg, Germany). Cells were propagated under standard incubation conditions with 5% atmospheric CO_2_ at 37 °C, as previously described [52,53].

Transient transfection of small interfering RNA (siRNA) duplexes against ADAR (Sigma-Aldrich, Hamburg, Germany#SASI_Hs01_00244017) (10 nM) was carried out using RNAiMax (Thermo Fischer Scientific, Darmstadt, Germany) according to the manufacturer’s protocol. The corresponding amount of control siRNA (MISSION Negative control SIC002, confidential sequence (Sigma-Aldrich, Hamburg, Germany) was used. Forty-eight hours after the transfection of siRNAs, cells were harvested using TRIzol (Thermo Fischer Scientific, Darmstadt, Germany) to extract RNA or lysed for protein analysis.

After the purification and treatment of total RNA with TURBO DNase (Thermo Fischer Scientific, Darmstadt, Germany) to digest genomic DNA, 1 μg of RNA was reverse transcribed with SuperScript VILO Master Mix (Thermo Fischer Scientific, Darmstadt, Germany). For quantitative reverse transcription polymerase chain reaction (qRT-PCR), 1 μL of the cDNA template was used with Fast SYBR Green Master Mix (Thermo Fischer Scientific) via StepOne Plus Real-Time PCR System (Applied Biosystem, Darmstadt, Germany) with the following thermal cycling condition: 95 °C for 20 sec followed by 40 cycles of 95 °C for 3 s and 60 °C for 30 s. Relative fold expression was calculated by 2^−ΔΔCt^ using glyceraldehyde 3-phosphate dehydrogenase (*GAPDH*) as an internal control. The list of primer sequences can be found in Appendix A. 

### 2.3. Western Blotting

Western blotting experiments were carried out via Expedeon Blot System (Expedeon, Cambridge, United Kingdom) using pre-cast 4%–12% gradient SDS-PAGE gel (Expedeon, Cambridge, United Kingdom). For each well, 50 μg of total extract were diluted in 20 μL of RunBlue LDS Sample Buffer (Expedeon, Cambridge, United Kingdom) with 0.1 M dithiothreitol (DTT) and loaded after boiling at 70 °C for 10 min. The samples were fractionated by migration under 180 volts for 70–90 min.

After the electrophoretic separation, gels were blotted in TGS Buffer 1X (0.025 M Tris, 0.192 M glycine, 10% methanol, and 0.1% SDS) using polyvinylidene fluoride (PVDF) membrane (Amersham, Munich, Germany) with a constant voltage of 200 volts at 4 °C for 90 min. After the transfer, the membrane was blocked with 5% skimmed milk diluted in tris-buffered saline (TBS) for 60 min. Then, the membrane was incubated with the following antibodies (diluted in 5% skimmed milk TBS) at 4 °C for overnight: anti-ADAR1 (Abcam, Cambridge, United Kingdom, #ab88574; diluted at 1:500) and anti-histone H3 (Abcam, #1791; diluted at 1:1000) as a loading control. After primary antibody incubation, the membrane was extensively washed with 5% skimmed milk diluted in TBS with 0.5% of Tween20 (TBS-T) and incubated with anti-rabbit-HRP (horseradish peroxidase; Roth; diluted at 1:1000) in 5% skimmed milk + TBS-T. After three 15 min washes with 5% milk and TBS-T, the membrane was revealed using Luminata Forte Western HRP substrate (Merck Millipore, Darmstadt, Germany) via C-DiGit Blot Scanner (LI-COR).

### 2.4. RNA Immunoprecipitation (RIP)

Magna RIP Kit (Millipore, Darmstadt, Germany) was used according to the manufacturer’s protocol. For each RIP reaction, 100 μL of cellular pellet from HEK-293 cells was fixed with 1% formaldehyde in PBS at room temperature for 10 min. Cross-linking reaction was stopped by adding 590 μL of 2.5 M glycine. Fixed cells were subsequently harvested and resuspended in RIP lysis buffer supplemented with protease/RNAse inhibitors. Lysate was obtained using a dounce homogenizer on ice (dounced 10 times for releasing nuclei) followed by incubation on ice for 15 min. An equal volume of RIP lysis buffer was added to the cellular pellet. From the solution, 10 μL (10%) of lysate was removed and stored as an “input”. For each RIP reaction, 100 μL of lysate was mixed with 5 μg of rabbit anti-IgG (immunoglobulin G; negative control provided with the kit), anti-HuR (Millipore, #03-102), and anti-AGO2 (Argonaute 2; Millipore, #03-110) antibodies previously conjugated with protein A/G magnetic beads (provided with the kit). After incubation at +4 °C overnight, RNA-protein immune complex was extensively washed with RIP Wash Buffer (provided with the kit). The cross-linking was reversed by incubation with proteinase K. The immune-precipitated RNA was purified through phenol/chloroform/isoamyl alcohol (5:1:1). The purified immuno-precipitated RNA was treated with DNase I and reverse transcribed using SuperScript VILO Master Mix.

### 2.5. RNA-seq

One microgram of DNase-I-treated total RNA was used for the strand-specific cDNA library construction at Novogene (Sacramento, CA, USA) using NEBNext Ultra Directional RNA Library Prep Kit for Illumina (New England BioLabs, #E7420L) according to the manufacturer’s protocol. Briefly, ribosomal RNA was removed using Ribo-Zero rRNA Removal kit (Illumina, San Diego, CA, USA, #MRZH11124), and fragmented randomly by adding fragmentation buffer. The first strand cDNA was synthesized using random hexamer primers. The second strand cDNA was generated incorporating deoxyuridine triphosphate (dUTP) in place of deoxythymidine triphosphate (dTTP) to create blunt-ended cDNA. After a series of terminal repair, poly-adenylation, and sequencing adaptor ligation, the double-stranded cDNA library was completed following size selection and PCR enrichment. The resulting 250–350 bp insert libraries were quantified using a Qubit 2.0 fluorometer (Thermo Fisher Scientific) and quantitative PCR. The size distribution was analyzed using an Agilent 2100 Bioanalyzer. Qualified libraries were sequenced on an Illumina HiSeq 4000 Platform using a paired-end 150 run (2 × 150 bases). A minimum of 40 million raw reads were generated from each library.

The RNA-seq data generated in this study were deposited in the Gene Expression Omnibus (GSE141095).

### 2.6. Statistics

Data are presented as mean ± standard error of the mean (S.E.M.). *F*-test was applied for normality. Two-sample, two-tailed Student’s *t*-test was performed to calculate a *p*-value via Microsoft Excel.

## 3. Results

### 3.1. Presence of RNA Editing Sites in RBP-Bound Regions

Given that CLIP-seq utilizes NGS as readout, in principle, it is possible to identify the presence of RNA editing sites within the sequences that RBPs bind. To test this possibility, the analyzed CLIP-seq data were downloaded from the starBase database [45,46]. Among all CLIP-seq data available in the starBase database, human embryonic kidney cells 293 (HEK-293) were chosen for further analysis because they constitute the largest number of analyzed data (48 out of 86 available datasets). In CLIP-seq data of HEK-293 cells, 27 RBP were targeted. Using the RBP-bound regions identified from CLIP-seq data (Appendix A), the presence of RNA editing sites was searched via the RADAR database [27]. When these sites were examined for their genic locations, numerous RNA editing sites were identified in the RBP-bound regions (Figure 1; Appendix A), although there were significant variations among the RBPs examined. These RNA editing sites can be found in both Alu (Appendix A) and non-Alu elements (Appendix A), as RBP recognize specific sequences. Given that 75.17% of the RNA editing sites in both Alu and non-Alu elements were found in the intronic regions followed by intergenic and 3’-UTR (20.06% and 3.31%, respectively) (Table 1), it was not surprising that many RBP-bound regions with RNA editing sites were found in these three regions.

The merged data above show large numbers of RNA editing sites in RBP-bound regions, which raises the question as to whether such RNA editing sites are real. Given that the RADAR database contains RNA editing sites from various cell types and tissues, it could be that the above approach overestimated the number of RNA editing sites present in RBP-bound regions. Furthermore, these RNA editing sites are not strictly from HEK-293 cells, in which RBP-bound regions were identified. To investigate this point, the published RNA-seq data of HEK-293 cells were analyzed using our RNAEditor [28], which resulted in the identification of 109,283 RNA editing sites. Indeed, fewer numbers of RNA editing sites were identified in RBP-bound regions in HEK-293 cells (Appendix A) compared to various cell types and tissues from the RADAR database (Appendix A). On the basis of these results, we concluded that the same cell type must be used for the analysis of RNA editing sites and their presence in RBP-bound regions.

### 3.2. Consequence of Loss of ADAR for the Binding of RBP

ADARs catalyze RNA deamination on the basis of their structures rather than having the defined sequence motif [54,55,56,57]. This fact makes it difficult to simply compare the sequence motifs of RBP to those of ADARs to derive RBP-binding sites to be edited. As we previously reported [28], it is rather common that RNA editing sites are present in clusters, which we named RNA editing islands [28]. As these RNA editing islands reflect the bound regions of ADAR, which were verified by RIP-seq data generated using anti-ADAR antibody [58], RNA editing islands could be viewed as more probable and important RNA editing sites. Given the situations discussed above, we examined the degree of overlaps between RNA editing islands and RBP-bound regions in HEK-293 cells. From the RNA-seq data of HEK-293 cells, 21,120 RNA editing islands were identified using our RNAEditor (Table 2; Appendix A). When these RNA editing islands were compared to the RBP-bound regions (Appendix A), HuR-bound regions possessed the greatest number of RNA editing islands, which is consistent with previous reports regarding ADAR-regulating gene expression and stability of messenger RNAs (mRNAs) via interaction with HuR, as in the case of cathepsin S (*CTSS*) [19].

Although the evidence above is informative, one cannot reliably conclude that RNA editing sites and islands have effects on RBP-bound regions. To investigate this point, we hypothesized that upon silencing of ADAR via siRNA, the number of RNA editing islands and the magnitude of RBP binding will be altered compared to the control condition (i.e., HEK-293 cells transfected with siRNA against random sequence) (Figure 2A). To test this hypothesis, we chose to knockdown *ADAR* (*ADAR1*), whose binding overlaps to RNA editing islands [28]. The siRNA against *ADAR* targets both isoforms of ADAR1, which are p110 and p150 (Figure 2B). For the RBP, HuR was chosen, as this RBP showed the greatest number of RNA editing sites and islands in its bound regions (Figure 1). RNA immunoprecipitation, followed by RT-PCR (RIP-PCR) experiment, was performed for six genes [*CYP20A1* (cytochrome P450, family 20, subfamily A, polypeptide 1), *GNL3L* (G protein nucleolar 3 like), *LYRM7* (LYR motif containing 7), *MAVS* (mitochondrial antiviral signaling protein), *PDDC1* (Parkinson disease 7 domain containing 1), and *TAF8* (TATA-box binding protein associated factor 8)] by using primer pairs targeting their 3’-UTR regions, which overlapped between the RNA editing islands and HuR-binding regions (Figure 2C). The result showed that significantly less binding of HuR to these genes was recorded for half of the genes (*CYP20A1*, *GNL3L*, and *LYRM7*), suggesting that, to a certain extent, the existence of RNA editing islands is necessary for the efficient binding of HuR. To investigate this finding further, RNA-seq experiment was performed upon silencing of ADAR in HEK-293 cells. When the RNA editing sites were compared for the six genes above, the decreased numbers of RNA editing sites were detected for *CYP20A1*, *GNL3L*, *LYRM7*, and *MAVS*, especially in 3’-UTR regions (Appendix A). However, no RNA editing sites were detected in 3’-UTR regions of *PDDC1* and *TAF8*, which suggest for no difference in binding of HuR, as shown in Figure 2C.

To further confirm the above findings, another RBP was tested. Given that the previous data targeted 3’-UTR regions, we chose to test for the alteration in binding of Argonaute protein, which is an essential component of RNA-induced silencing complex (RISC). As shown in Figure 2D, silencing of *ADAR* affected the binding of argonaute RISC catalytic component 2 (AGO2). As *ADAR* was knocked down, it could be possible that the expressions of these genes were reduced, which caused alterations in binding of RBPs to these genes. However, this was not the case, as all genes showed no statistically significant difference in their expressions between the control and silencing of *ADAR* (Figure 2E). These data confirm that reduction of RNA editing islands caused by silencing of *ADAR* affects the binding of RBPs.

## 4. Discussion

In this study, the extent to which RNA editing events affect RBP binding was investigated. Building upon our recently introduced RNAEditor bioinformatics tool to detect RNA editing sites and islands from RNA-seq data, we examined the overlap between RNA editing events and RBP-bound regions. Here, we experimentally validated the necessity of ADAR RNA editing enzyme to the binding of the RBP HuR in HEK-293 cells. The major findings of this study were (1) many RNA editing sites can be found in the regions that RBPs bind to; (2) RNA editing islands can be used to investigate the influence of RNA editing events to RBP-bound regions; and (3) experimentally, the lack of RNA editing islands upon silencing of RNA editing enzyme ADAR reduces HuR binding compared to the control.

Utilizing CLIP-seq data in HEK-293 cells, we examined 27 RBP and their binding to RNA editing sites. Among these RBPs, by far, HuR (encoded by *ELAVL1* gene) bound the most often to RNA editing sites. Given that HuR stabilizes RNAs to regulate gene expression [59,60,61], it is tempting to speculate that RNA editing sites promote the binding of HuR to stabilize the edited transcripts, as we have shown in our previous study for cathepsin S (*CTSS*) in endothelial cells [19]. Although an earlier study indicated such a mechanism of action [58], two other studies that investigated this interaction between RNA editing and HuR suggest the opposite [62,63]. These studies report that HuR, as a de-stabilizing enzyme influenced by the presence of RNA editing sites in the target RNA, nuclear-retained Cat2-transcribed nuclear RNA (*Ctn* RNA) [62,63]. As we did not investigate the stability of RNA in this study, we lack the evidence to refute these previous two studies. In order to investigate this point, RNA stability must be measured upon silencing of ADAR by treating the cells with actinomycin D to suppress the synthesis of mRNAs.

Over three-quarters of the RNA editing sites are found in the intronic regions. Thus, it is more likely that RNA editing affects splicing patterns. It is interesting to note that identifying RNA editing sites in spliced exons and junctions is challenging, as the sequencing reads could be mapped beyond the corresponding exon boundaries [6,15,17,28]. Thus, RNA editing sites near the splicing acceptor and donor sites cannot be accurately identified unless the direct comparison of RNA-seq data to the whole-genome sequencing (WGS) from the same cell is conducted, which is not ideal as the cost of WGS is much more (including computational time) than simply performing bulk RNA-seq. Thus, it is possible that the effect of RNA editing to splicing patterns is underestimated. An additional problem is that it is still a challenge to correctly identify spliced exons from short-read RNA-seq (e.g., Illumina sequencer) as ≈95% of multi-exon genes undergo alternative splicing, which results in ≈100,000 alternative splicing events in major human tissues [64]. One potential solution is to use “Iso-Seq” [65,66,67,68,69,70,71,72,73,74,75,76], which allows for the capturing of transcripts over kilo nucleotides (nt) in length, recording isoforms with increased accuracy compared to shorter sequences (e.g., <200 nt). However, the analysis of long reads is difficult due to the low sequence coverage, which also leads to missing of RNA editing sites to be identified. In order to understand the effects of RNA editing, a combined method of different types of RNA-seq and WGS along with bioinformatics analysis is urgently needed.

## Figures and Tables

**Figure 1 high-throughput-08-00019-f001:**
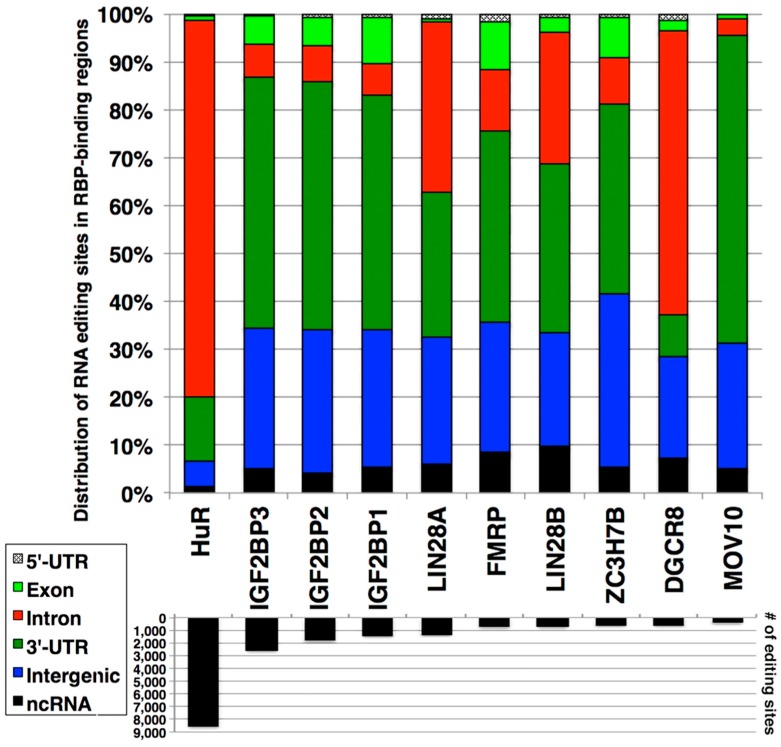
Overlap between RNA editing sites and RNA-binding protein (RBP)-bound regions. The upper graph shows the genic distribution of the overlapped sites. The lower graph indicates the number of overlapping RNA editing sites. The top 10 overlapping RBPs are shown. 5′-untranslated region (5′-UTR); 5′-untranslated region (3′-UTR); non-coding RNA (ncRNA).

**Figure 2 high-throughput-08-00019-f002:**
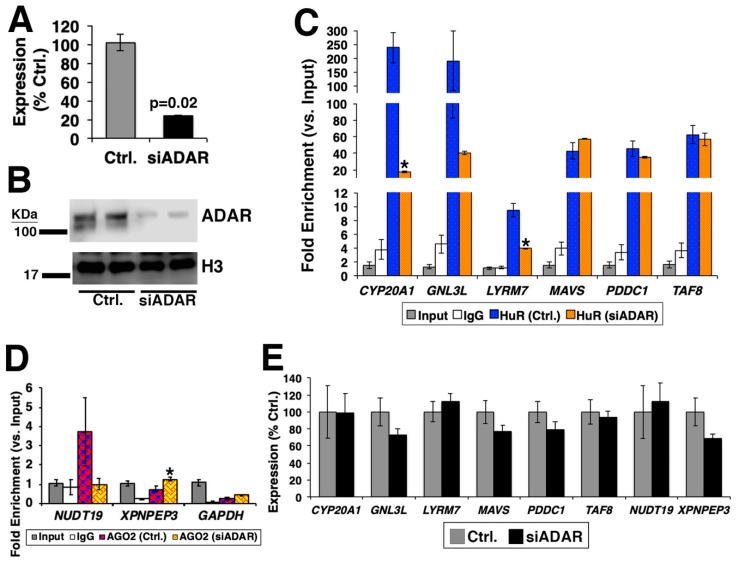
Knockdown of adenosine deaminases acting on RNA (ADAR) in HEK-293 cells. (**A**) qRT-PCR results of ADAR expression upon silencing of ADAR (siADAR). HEK-293 cells transfected with siRNA against random sequence was used as control (Ctrl.). *n* = 3 technical replicates. (**B**) Western blotting results of normally grown HEK-293 cells (normal), control, and silencing of ADAR. Antibodies against ADAR and histone H3 (as a loading control) were used. *n* = 2 technical replicates. (**C**,**D**) RNA immunoprecipitation (RIP)-PCR. *n* = 3 technical replicates. The fold enrichment was calculated against the average Ct value of the input. * represents *p* < 0.05 comparing (**C**) HuR- and (**D**) AGO2-binding between silencing of ADAR and control. (E) Expressions of genes. *n* = 3 technical replicates. No statistically significant difference between silencing of ADAR and control was recorded in all genes.

**Table 1 high-throughput-08-00019-t001:** Distribution of genic locations of RNA editing sites listed in the RADAR database.

Genic Location	All Sites	Alu	Non-Alu
5′-UTR	6775	6131	644
Exon	4405	2082	2323
Intron	1,936,801	1,870,644	66,157
3′-UTR	85,169	79,434	5735
Intergenic	516,714	479,480	37,234
ncRNA	26,595	24,184	2411

**Table 2 high-throughput-08-00019-t002:** Distribution of genic locations of RNA editing islands of HEK-293 cells.

Genic Location	Count
5′-UTR	50
5′-UTR;Intron	10
Exon	20
Exon;Intron	9
Intron;Exon	17
Intron;5′-UTR	4
Intron	16,707
Intron;3′-UTR	14
Intron;ncRNA	33
3′-UTR	590
3′-UTR;Intergenic	2
3′-UTR;Intron	4
Intergenic;5′-UTR	1
Intergenic	3376
Intergenic;3′-UTR	2
Intergenic;ncRNA	3
ncRNA	258
ncRNA;Intron	18
ncRNA;Intergenic	2

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
