# Peer review of "Investigation of RNA Editing Sites within Bound Regions of RNA-Binding Proteins"

_2571-5135, 2019, doi:10.3390/ht8040019_

Round 1
Reviewer 1 Report
Through employing bioinformatic techniques, this study reports that RNA editing sites occur frequently in RBP-bound regions. It also showed that the binding of one RBP, HuR, was reduced upon silencing of the RNA editing enzyme ADAR. The possibility that the presence of RNA editing islands influences RBP binding to its target regions suggests that RNA editing may act as an important mediator of RBP-RNA interactions.
The manuscript could be improved by addressing the following concerns:
The authors should specify which ADAR is under study, ADAR1-p150, ADAR1-p110, or ADAR2. It would be helpful to the readers if the authors could provide some examples of transcript folding/structural schematic that show the correlated occurrence of RNA editing sites/clusters and RBP target sequences.Author Response
Thank you very much for the valuable comments regarding our manuscript.
Specific responses to each of the reviewers’ comments (in italicized font) are listed below and are followed by our response (in bold font):
Response to Reviewer #1:
Through employing bioinformatic techniques, this study reports that RNA editing sites occur frequently in RBP-bound regions. It also showed that the binding of one RBP, HuR, was reduced upon silencing of the RNA editing enzyme ADAR. The possibility that the presence of RNA editing islands influences RBP binding to its target regions suggests that RNA editing may act as an important mediator of RBP-RNA interactions.
The manuscript could be improved by addressing the following concerns:
The authors should specify which ADAR is under study, ADAR1-p150, ADAR1-p110, or ADAR2.
Response: Thank you very much for your praise. We studied ADAR1. The antibody that we employed detects both isoforms of ADAR1 (p110 and p150). These facts are now clearly stated in the manuscript as follows:
“To test this hypothesis, we chose to knockdown ADAR (ADAR1), whose binding overlap to RNA editing islands [28]. The siRNA against ADAR employed targets both isoforms of ADAR1, which are p110 and p150 (Figure 2B).”
It would be helpful to the readers if the authors could provide some examples of transcript folding/structural schematic that show the correlated occurrence of RNA editing sites/clusters and RBP target sequences.
Response: This is an interesting point to be considered. However, the transcript folding and structures are difficult to study and predict as these processes are dynamic as in the case of protein folding. There are biochemical methods available, such as FragSeq and PARS, although it is still technically challenging to obtain consistent results. We do not want to provide the readers with any ambiguity without showing firm experimental data about how silencing of ADAR affects transcript folding and structures. It would take further experiments with more advanced techniques to solve this point, which is well beyond the scope of this manuscript.
Reviewer 2 Report
In this manuscript, Weirick et al. presented a study that correlates RNA editing sites and RBP binding sites, and also experimentally investigated how knockdown of ADAR may affect HuR binding to RNAs in vivo. Although interesting correlations have been observed, the study seems to be largely incomplete.
Major comments:
[1] The authors aimed to investigate how RNA editing events affects of the binding of RBPs, but the correlation analysis presented in Figure 1 doesn’t provide insights whether RNA editing events affect RBP binding. Even though the identified sites are correlated between RNA editing and specific RBP binding, it is possible that there isn’t any connection between the two and it is just randomized signal overlapping. A statistical analysis is necessary for the significance of the correlation.
[2] Since RNA editing frequently occurs at dsRNA structures, how does the listed RBPs prefers dsRNA structures? Are the common sites just an indication of sharing RNA identity elements?
[3] Furthermore, how is the RBP binding affecting RNA editing? For example, HuR sitting on the editing site would provide steric hindrance for ADAR binding. Therefore, the author also needs to investigate the other side of the story about how RBP binding affects ADAR at specific editing sites experimentally.
[4] Presenting the correlation as tables in the form of number of sites at various region of the RNA is not very informative. Metaplots with editing frequency and RBP binding enrichment could be much more helpful and informative. For example, not all sites are edited the same way and not all CLIP sites are bound by the RBP the same extend.
[5] The analysis pipeline/method used to identify RNA editing sites in RBP binding regions wasn’t clearly described. There is only one sentence in 2.1 regarding using the BEDTools and -wo options. Other parameters and applied thresholds are needed.
[6] In Figure 2, data is needed for examination of the effects of ADAR knockdown on editing islands in the six genes (Fig 2C). If the hypothesis is correct, editing efficiency might only go down in the first three RNAs but not the last three RNAs, to explain the results that HuR binding efficiency was only affected in the first three RNAs.
[7] Discussion of the previous work (ref 19 and 58) on RNA editing and HuR binding needs to be more extensive to explain the novelty of this work.
Author Response
Thank you very much for the valuable comments regarding our manuscript.
Specific responses to each of the reviewers’ comments (in italicized font) are listed below and are followed by our response (in bold font):
Response to Reviewer #2:
In this manuscript, Weirick et al. presented a study that correlates RNA editing sites and RBP binding sites, and also experimentally investigated how knockdown of ADAR may affect HuR binding to RNAs in vivo. Although interesting correlations have been observed, the study seems to be largely incomplete.
Major comments:
[1] The authors aimed to investigate how RNA editing events affects of the binding of RBPs, but the correlation analysis presented in Figure 1 doesn’t provide insights whether RNA editing events affect RBP binding. Even though the identified sites are correlated between RNA editing and specific RBP binding, it is possible that there isn’t any connection between the two and it is just randomized signal overlapping. A statistical analysis is necessary for the significance of the correlation.
Response: Thank you very much for critically analyzing our manuscript. This is the exact point that we would like to raise as a question to the field and answer to a certain extent as we provide in this manuscript. Figure 1 is to demonstrate that there are RNA editing sites present in CLIP-seq data, which were performed in various laboratories and summarized in the starBase database. In order to avoid any biases, we did not modify the data provided by the starBase database; we simply downloaded the data as they are provided. The same applies to RNA editing sites provided by the RADAR database. We would like to emphasize that these are databases that have been used by researchers around the world and cited many times, which make them as highly accessed databases for various researches. Since there are several thousand sites that each RBP bind and most of such data are provided with n=1 as shown in Table S2, it is not possible to reliably apply any statistical analysis as it would be very challenging to normalize such binding sites and the presence of RNA editing sites as well as having a negative control for such data. Since there is no consensus in the field as of now for whether RNA editing sites affect the binding of RBPs, our study provides an evidence that there is a possible correlation between the presence of RNA editing sites and the binding of RBPs. To confirm this possible correlation and relationship, we performed RIP-PCR upon silencing of ADAR in Figure 2.
[2] Since RNA editing frequently occurs at dsRNA structures, how does the listed RBPs prefers dsRNA structures? Are the common sites just an indication of sharing RNA identity elements?
[3] Furthermore, how is the RBP binding affecting RNA editing? For example, HuR sitting on the editing site would provide steric hindrance for ADAR binding. Therefore, the author also needs to investigate the other side of the story about how RBP binding affects ADAR at specific editing sites experimentally.
Response: It is true that ADARs bind dsRNA to perform A-to-I conversion. Once such conversion is made, it destabilizes dsRNA because A:U base pairs are disrupted than I:U mismatches [PMID: 28362255]. We do not speculate (or hypothesize) that the edited RNA remains as double-stranded as most of RBPs that own RRM domains preferentially bind single-stranded RNA as written in our Introduction. Thus, we hypothesize the sequential cascade of events take place; that are: 1) RNA is transcribed from the genomic DNA; 2) The transcribed RNA is modified, including RNA editing; and 3) The modified RNA is bound by RBPs for further processing, including splicing, mRNA transport (from a nucleus to cytoplasm) ,and translation.
[4] Presenting the correlation as tables in the form of number of sites at various region of the RNA is not very informative. Metaplots with editing frequency and RBP binding enrichment could be much more helpful and informative. For example, not all sites are edited the same way and not all CLIP sites are bound by the RBP the same extend.
Response: We provide supplementary tables to the readers so that the whole study can be repeated and validated by the readers as the original data are provided by the starBase and RADAR databases. We did not calculate editing frequency for each bound region of RBP as the original experiments (those collected in the starBase and RADAR databases) were performed in various laboratories around the world. Thus, there are many operator-dependent biases present in each data. Since both starBase and RADAR databases have been used by numerous researchers around the world and cited accordingly, our primary motive in this manuscript is to combine these two data sources (without any modifications to the analyzed data provided) to examine to which extend such datasets overlap from the standpoint of RNA editing sites and RBP-bound regions, which are summarized in Figure 1.
[5] The analysis pipeline/method used to identify RNA editing sites in RBP binding regions wasn’t clearly described. There is only one sentence in 2.1 regarding using the BEDTools and -wo options. Other parameters and applied thresholds are needed.
Response: There is no modification made to the original data provided by the starBase and RADAR databases. As such, we did not apply any threshold nor change any parameters to make the data fit to a certain criterion. The data were downloaded from the above two databases and simply compared using BEDTools as stated in the subsection 2.1.
[6] In Figure 2, data is needed for examination of the effects of ADAR knockdown on editing islands in the six genes (Fig 2C). If the hypothesis is correct, editing efficiency might only go down in the first three RNAs but not the last three RNAs, to explain the results that HuR binding efficiency was only affected in the first three RNAs.
Response: The experiments performed in Figure 2 are to confirm the efficiency of HuR binding to RNA editing islands identified from the published RNA-seq data of HEK-293 cells. It is possible to check for the reduction of RNA editing sites by performing RNA-seq experiment upon silencing of ADAR in our experimental setting using HEK-293 cells. However, doing such experiment only provides an additional proof that silencing of ADAR reduces the number of RNA editing sites. As shown in Figure 1, we screened for the possible overlap between RNA editing sites and RBP-bound regions. The experiments formed in Figure 2 are to test whether the binding of one of such RBPs, HuR, will be affected by the reduced number of RNA editing sites within the HuR-bound regions.
[7] Discussion of the previous work (ref 19 and 58) on RNA editing and HuR binding needs to be more extensive to explain the novelty of this work.
Response: The above sentence was modified as follow:
“When these RNA editing islands were compared to the RBP-bound regions (Table S8), HuR-bound regions possessed the greatest number of RNA editing islands, which is consistent with previous reports regarding ADAR regulating gene expression and stability of mRNAs via interaction with HuR as in the case of cathepsin S (CTSS)] [58].”
Reviewer 3 Report
The manuscript describes an application of an earlier published bioinformatic tool, which predicts potential RNA editing sites from the sequence. The paper shows a correlation between RNA editing sites and regions where RNA-binding proteins (RBPs) can interact. The authors then present the results of experiments where binding of an RBP, HuR, is detected both in the presence and absence of ADAR expression.
Mechanically, the paper is fairly well written. There are quite a few grammatical/format issues that are scattered throughout the paper that should be corrected. For example, in the introduction, the extensive use of quotation marks is not warranted (pg 2, middle paragraphs). The sentence spanning lines 76-77 should be corrected to “…it is generally categorized as an RNA-binding protein (RBP)”. There are others like this as well. Generally, these are pretty minor errors, but a couple more rounds of editing would be beneficial to make the manuscript a little more readable.
Other minor concerns:
The title, as written, doesn’t make sense. Maybe something like “Correlation” instead of “Systematic investigation” would be better.
Line 80: The abbreviation NGS should be written out the first time it is used.
Line 97: “alternations” should be corrected to “alterations”
Line 108: A little more background might be helpful. What is the significance of mapping the data sets to hg19?
Line 116: insert “and” at the beginning of the line
Line 202: insert “deamination” after “RNA”
Figure 2D: The last two sets of data, NUDT19 and XPNPEP3, are not discussed anywhere in the manuscript.
Major concerns:
There are many different ADAR enzymes. Which one was used, and why was it chosen over the others for the purposes of the study? Has the binding of HuR to RNA been shown to be sequence-specific, maybe by biophysical methods? If so, citing these references would greatly strengthen the claim that there is a correlation between the ADAR silencing and HuR binding. Looking at Figure 2C, it would seem that silencing of ADAR indeed has a large effect on HuR binding to RNA for CYP20A1 and LYRM7. Even though not statistically significant, the GNL3L gene looks like it may be affected too. However, the other 3 genes show little to no change upon ADAR silencing. To demonstrate the utility of this approach, the authors should, at minimum, speculate as to why some genes are affected and not others. Only one protein (HuR) was examined for its RNA binding with and without ADAR editing. The manuscript would be much more convincing if this effect were demonstrated for other RBPs as well.
In summary, the paper describes a very interesting and potentially useful tool for the quick and accurate prediction of RNA editing by ADARs. In order to be truly useful and of interest to the scientific community, the authors either need to perform more experiments on other RBPs and/or present a more convincing explanation of why the ADAR silencing effect was not universal.
Author Response
Thank you very much for the valuable comments regarding our manuscript.
Specific responses to each of the reviewers’ comments (in italicized font) are listed below and are followed by our response (in bold font):
Response to Reviewer #3:
The manuscript describes an application of an earlier published bioinformatic tool, which predicts potential RNA editing sites from the sequence. The paper shows a correlation between RNA editing sites and regions where RNA-binding proteins (RBPs) can interact. The authors then present the results of experiments where binding of an RBP, HuR, is detected both in the presence and absence of ADAR expression.
Mechanically, the paper is fairly well written. There are quite a few grammatical/format issues that are scattered throughout the paper that should be corrected. For example, in the introduction, the extensive use of quotation marks is not warranted (pg 2, middle paragraphs). The sentence spanning lines 76-77 should be corrected to “…it is generally categorized as an RNA-binding protein (RBP)”. There are others like this as well. Generally, these are pretty minor errors, but a couple more rounds of editing would be beneficial to make the manuscript a little more readable.
Response: Thank you very much for carefully reading our manuscript. The above grammatical errors and other errors have been corrected accordingly.
Other minor concerns:
The title, as written, doesn’t make sense. Maybe something like “Correlation” instead of “Systematic investigation” would be better.
Response: The title is now changed to: RNA editing correlates to the binding of RNA-binding proteins.
Line 80: The abbreviation NGS should be written out the first time it is used.
Line 97: “alternations” should be corrected to “alterations”
Response: The above points have been corrected accordingly.
Line 108: A little more background might be helpful. What is the significance of mapping the data sets to hg19?
Response: The following sentence has been added:
“All of the above data sets were originally mapped to hg19. In order to avoid any biases resulting from further secondary data analysis, the data were downloaded from the above databases without any modifications.”
Line 116: insert “and” at the beginning of the line
Line 202: insert “deamination” after “RNA”
Response: We were unable to locate the exact positions for the above changes. Could you kindly provide more information?
Figure 2D: The last two sets of data, NUDT19 and XPNPEP3, are not discussed anywhere in the manuscript.
Response: Thank you very much for noticing this error. Figure 2 modified accordingly.
Major concerns:
There are many different ADAR enzymes. Which one was used, and why was it chosen over the others for the purposes of the study?
Response: Thank you very much for carefully reading our manuscript. We studied ADAR1 (official gene name, ADAR). The antibody that we employed detects both isoforms of ADAR1 (p110 and p150). These facts are now clearly stated in the manuscript as follow:
“To test this hypothesis, we chose to knockdown ADAR (ADAR1), whose binding overlap to RNA editing islands [28]. The siRNA against ADAR employed targets both isoforms of ADAR1, which are p110 and p150 (Figure 2B).”
Has the binding of HuR to RNA been shown to be sequence-specific, maybe by biophysical methods? If so, citing these references would greatly strengthen the claim that there is a correlation between the ADAR silencing and HuR binding.
Response: The following sentence was modified to reflect the above point:
“When these RNA editing islands were compared to the RBP-bound regions (Table S8), HuR-bound regions possessed the greatest number of RNA editing islands, which is consistent with previous reports regarding ADAR regulating gene expression and stability of mRNAs via interaction with HuR as in the case of cathepsin S (CTSS)] [58].”
Looking at Figure 2C, it would seem that silencing of ADAR indeed has a large effect on HuR binding to RNA for CYP20A1 and LYRM7. Even though not statistically significant, the GNL3L gene looks like it may be affected too. However, the other 3 genes show little to no change upon ADAR silencing. To demonstrate the utility of this approach, the authors should, at minimum, speculate as to why some genes are affected and not others. Only one protein (HuR) was examined for its RNA binding with and without ADAR editing. The manuscript would be much more convincing if this effect were demonstrated for other RBPs as well.
In summary, the paper describes a very interesting and potentially useful tool for the quick and accurate prediction of RNA editing by ADARs. In order to be truly useful and of interest to the scientific community, the authors either need to perform more experiments on other RBPs and/or present a more convincing explanation of why the ADAR silencing effect was not universal.
Response: We now provide an additional data using anti-AGO2 antibody in Figure 2. Given that we examined 3’-UTR regions that overlap between the RNA editing islands and HuR-binding regions, targeting another RBP, Argonaute, would further strengthen our manuscript.
Round 2
Reviewer 2 Report
Confirming the correlation remains the main issue of the manuscript. The high number of editing event in the HuR looks very promising, but the type of analysis the authors was using (mostly presented in the supplemental files) is just identifying number of editing sites in RBP binding sites, far from a true validation of a positive correlation.
In the authors' response to major comment 1, the authors stated that "it is not possible to reliably apply any statistical analysis as it would be very challenging to normalize such binding sites and the presence of RNA editing sites ". While I understand the technical difficulty in doing such analysis using the downloaded data, no statistical correlation analysis implies that there might not be much correlation. To make it simple, the author can simply analyze the following: a) the frequency of editing event across the genome, and b) the frequency of editing event at each RBP binding site. Is b>a for all the RBPs listed in Figure 1? Without such information, correlation remains a question.
The major comment 6 still needs to be addressed experimentally. The extra experiment of examination of the effects of ADAR knockdown on editing islands in the six genes (Fig 2C) is necessary to support that the change in HuR binding specfically for the three genes is truly caused by diminishing RNA editing. The authors' response that "doing such experiment only provides an additional proof that silencing of ADAR reduces the number of RNA editing sites" is misleading.
Author Response
There may be really a misunderstanding of the manuscript. We did further analysis of the data and updated Table S3 as we wrote in our response in last revision as well as changing the title.
Concerning the major point 1, we would like to clarify that we simply accepted the suggestion of Reviewer #3 to change the title to include the word "correlates". In the main text, we did not use any words related to "correlation" between RBP-bound regions and RNA editing sites. In the beginning of the Results section, we wrote as follows:
"3.1. Presence of RNA editing sites in RBP-bound regions
Given that CLIP-seq utilizes NGS as readout, in principle, it is possible to identify the presence of RNA editing sites within the sequences that RBPs bind. To test this possibility, the analyzed CLIP-seq data were downloaded from the starBase database [45,46]. Among all CLIP-seq data available in the starBase database, human embryonic kidney cells 293 (HEK-293) were chosen for further analysis because they constitute the largest number of analyzed data (48 out of 86 available data sets). In CLIP-seq data of HEK-293 cells, 27 RBP were targeted. Using the RBP-bound regions identified from CLIP-seq data (Table S2), the presence of RNA editing sites was searched via the RADAR database [27]. When these sites were examined for their genic locations, numerous RNA editing sites were identified in the RBP-bound regions (Figure 1; Table S3), although there were significant variations among RBPs examined."
In short, we did not investigate the actual correlation between RBP-bound regions and RNA editing sites; rather, we searched for the presence of RNA editing sites within RBP-bound regions, regardless of enrichment of RNA editing sites compared to the whole transcriptome. To avoid any confusion, we changed the title to: "RNA editing sites influence the binding of RNA-binding proteins".
Furthermore, we updated Table S3 to include an additional information as follow: "To derive the degree of presence of RNA editing sites within RBP-bound regions ("% Presence"), the number of total edited sites within RBP-bound regions was divided by the number of bound regions shown in Table S2 for each RBP."
In summary, we simply investigated the presence (not the correlation) of RNA editing sites within RBP-bound regions.
Regarding the major comment 6, there is a misunderstanding. These RNA editing islands within six genes were chosen based on the presence of clusters (i.e., RNA editing islands) of RNA editing sites from RNA-seq data of HEK-293 cells that we analyzed using our RNAEditor. This part is written in details in the Materials and Methods section as follow: "To identify RNA editing sites and islands, the following RNA-seq data of HEK-293 cells were analyzed with RNAEditor [28]: SRR3994124, SRR3994125, SRR3994126, SRR629569 and SRR629570 [all are from the Sequence Read Archive (SRA)]."
The primer pairs used to detected these six genes are designed specifically to detect the RNA editing islands that were identified from the above RNA-seq data. In other words, RIP-PCR assay specifically targets the RNA editing island in each of six gene.
Reviewer 3 Report
The manuscript has been edited for grammatical errors and reads much better now. The two insertions that were mentioned in the first review are: the addition of "and" at the beginning of line 121 in the newest draft (so the line reads "and antibiotics."), and the addition of "deamination" to line 213 so that it reads "ADARs catalyze RNA deamination based on..."
The authors have sufficiently addressed my previous concerns regarding the generalizability of their assay by providing several more references that support their case. They have also clarified other points of confusion.
Author Response
Thank you very much for clarifying these two points. These points were corrected as suggested.
Round 3
Reviewer 2 Report
Thanks for the response. The title is still problematic especially for the word "influence" because as the the authors replied that they "simply investigated the presence (not the correlation) of RNA editing sites within RBP-bound regions." Maybe a good title should be "Investigation of the presence of RNA editing sites within RBP-bound regions".
However, this downgraded conclusion won't have the novelty and general interest to the audience.